# Air Quality Index as a Predictor of Respiratory Morbidity in At-Risk Populations

**DOI:** 10.3390/ijerph22101493

**Published:** 2025-09-27

**Authors:** Brandy M. Byrwa-Hill, Tricia Morphew, John O’Neill, Deborah Gentile

**Affiliations:** 1Department of Public Health, University of Utah School of Medicine, Salt Lake City, UT 84132, USA; 2Johns Hopkins Bloomberg School of Public Health, John Hopkins University, Baltimore, MD 21218, USA; tricia@morphewconsulting.com; 3Morphew Consulting, LLC, Bothell, WA 98201, USA; 4Department of Medical Affairs, Community Partners in Asthma Care, Canonsburg, PA 15317, USA; dgentile@francis.edu; 5Department of Emergency Medicine, Allegheny General Hospital, Pittsburgh, PA 15212, USA; john.oneill@ahn.org

**Keywords:** air quality index (AQI), asthma, bronchitis, PM_2.5_, SO_2_

## Abstract

The Mon Valley near Pittsburgh, Pennsylvania, consistently reports some of the poorest air quality in the United States. Recent studies have linked air pollution in this region to poor asthma outcomes but did not examine the impact on other respiratory conditions or vulnerable populations. This retrospective study examined the relationship between the air quality index (AQI) and respiratory exacerbations of asthma, bronchitis, and chronic obstructive pulmonary disease (COPD) in the Mon Valley between January 2018 and February 2020. We linked daily Air Quality Index (AQI) values for ozone, PM_2.5_, SO_2_ and NO_2_, plus temperature and wind speed to healthcare utilization for these conditions. Using a Poisson generalized linear model, we quantified the association between pollutant levels and same-day exacerbation rates, stratified analyses by age, sex, and insurance type to identify vulnerable subgroups. Results indicated that higher AQI scores, driven primarily by PM_2.5_ and SO_2_, were significantly associated with increased asthma exacerbations on the day of exposure. Children and individuals with public insurance experienced the greatest impact. Bronchitis exacerbations showed a delayed response to SO_2_. Our findings affirm PM_2.5_ and SO_2_ as key drivers of acute asthma events in the Mon Valley and extend this observation to include impacts on bronchitis and vulnerable populations. They also demonstrate the AQI’s value for public health surveillance and underscore the importance of tailored interventions such as issuing timely air quality alerts, strengthening emissions regulations, and improving access to preventive care to protect at-risk populations from adverse air pollution effects.

## 1. Introduction

The air quality index (AQI) serves as a useful tool for air quality monitoring and public health advisories in the United States and plays a central role in protecting populations from the adverse effects of air pollution on respiratory health [1]. Originally developed in the United States to provide air quality monitoring and public health advisories, the Air Quality Index (AQI) has since been widely adopted or adapted internationally as a key tool for communicating air pollution risks to the public [2]. Research has consistently demonstrated that poor air quality contributes to the exacerbation of respiratory symptoms, increased healthcare utilization, and a decline in lung function among vulnerable populations [3,4,5,6].

The significance of the AQI extends beyond individual health impacts, influencing environmental policy and public awareness at the national scale [7,8]. The Mon Valley region in Pennsylvania, encompassing parts of Allegheny County (including Pittsburgh), represents a relevant region for studying the respiratory health impacts of air pollution. Historically, dominated by industrial activities such as steel mills and coal-fired power plants, the Mon Valley is home to U.S. Steel’s Clairton Coke Works, the largest coke producing facility in North America, which has been identified as the largest industrial source of SO_2_ in the region [9]. Consequently, the Mon Valley consistently reports some of the worst air quality in the United States, especially in terms of elevated particulate matter (PM_2.5_) and sulfur dioxide (SO_2_) levels, which are pollutants known to exacerbate respiratory conditions. This legacy dates back to the 1948 Donora smog event, one of the deadliest industrial pollution episodes in the world which led to dozens of deaths and thousands of illnesses in the Mon Valley, this was a pivotal event in the history of U.S. air quality regulation [10].

Multiple studies underscore the importance of the Mon Valley in air quality research [11,12,13,14]. The 2018 Clairton Coke Works fire, for example, revealed a significant association between higher levels of SO_2_ and PM_2.5_ and a corresponding spike in outpatient visits, emergency department (ED) visits, and hospital admissions for asthma, illustrating the immediate effects of acute pollution events [7]. Other findings have revealed a strong correlation between short-term spikes in SO_2_ levels and increased asthma hospitalizations, particularly among children and older adults, emphasizing the importance of monitoring and controlling industrial emissions [15]. Longitudinal studies have shown that chronic exposure to high levels of PM_2.5_ is associated with both asthma incidence and severity [11,12,13,14]. A 2020 study highlighted delayed respiratory effects following pollutant exposure, whereas a 2021 study revealed significant increases in asthma attacks on days with relatively high PM_2.5_ and SO_2_ levels [16]. Socioeconomic disadvantage further amplifies Mon Valley’s vulnerability to air pollution. A 2015 study confirmed that lower-income communities often face higher exposure levels and greater health burdens, reinforcing the need for targeted public health interventions [17]. Additionally, they highlighted that lower-income communities often face higher exposure levels and greater health burdens, reinforcing the need for targeted public health interventions [17]. This aligns with the broader literature indicating that socioeconomic status (SES) plays a critical role in shaping health outcomes related to air pollution [15]. Moreover, regulatory efforts aimed at improving air quality have been shown to significantly reduce the incidence of respiratory diseases, including asthma exacerbations [16,18]. Finally, the global burden of disease suggests that individuals with chronic respiratory diseases such as chronic obstructive pulmonary disorder (COPD) face a disproportionately increased burden from ambient pollution [14,15].

While prior studies have linked air pollution in this region to asthma exacerbations, our study extends this work by examining a broader set of outcomes, three highly relevant respiratory conditions including asthma, COPD, and bronchitis, across both age and insurance groups. This approach allows us to assess not only the overall health impacts of unhealthy air quality but also differential effects among vulnerable populations, thereby providing new insights into health disparities within a region of persistently elevated exposures.

Our research aims to evaluate the disproportionate impact of air pollution across demographic groups in the Mon Valley by analyzing healthcare utilization data to assess exacerbation rates for respiratory conditions. The data from this study may be utilized to inform individual protective behaviors and support the development of targeted public policies aimed at improving air quality and reducing health disparities. This information can guide individuals in adopting protective measures and assist policymakers in developing regulations to improve air quality and reduce health risks.

## 2. Methods

This study was approved with a waiver of informed consent by the institutional review board (IRB) at the Allegheny Health Network Research Institute. The study used deidentified data retrospectively collected from electronic medical records (EMRs) (N = 117,545) of adults and children who resided within the Mon Valley, located within Allegheny County, Pennsylvania, an urban-industrial region of nearly 1.2 million residents. The Monongahela Valley (“Mon Valley”), within the county, is a historically industrialized corridor where valley topography and frequent temperature inversions contribute to elevated concentrations of fine particulate matter and sulfur dioxide (see Table A1 for a complete list of zip codes), and (Figure A1 for a map of the study location). Study participants were seen at a geographically convenient Allegheny Health Network (AHN) facility for acute exacerbations of respiratory disease between January 2018 and February 2020. Exacerbations were categorized by visit type (acute outpatient visit to primary or urgent care, acute visit to the ED, hospital preadmission observation, or hospital admission) with the most severe recorded on a given date for patient and diagnosis (asthma, bronchitis, or COPD) using their respective ICD-10 codes (Discharge diagnoses include acute asthma exacerbation (J45.901, J45.21, J45.31, J45.41, J45.51), status asthmaticus (J45.902, J45.22, J45.32, J45.42, J42.52), COPD exacerbation (J44.1), and acute bronchitis (J20.9). Specifically, the number of exacerbations was described in terms of numbers and percentages by sex, insurance type (private vs. public), and age groups (<18 years, 18–64 years, and ≥65 years) over a 790-day period. The American Community Survey (ACS) 5-Year Estimates (2015–2019) were used to characterize the study region’s overall demographics and to provide denominators for rate calculations (Table A1).

Air quality data was obtained from Environmental Protection Agency (EPA) regulatory monitoring sites selected based on their proximity to the study population region: Lawrenceville for ozone, Liberty for PM_2.5_, Braddock, and Liberty for SO_2_, and Lawrenceville and Parkway East for NO_2_. PM_10_ data were not analyzed because there have been no elevations of this criterion pollutant in the study region for several decades [19]. The AQI range equivalents for each pollutant were defined in accordance with the EPA designations for good, moderate, and unhealthy sensitive groups or those that are worse at the time of study: ozone 8 h (ppm): ≤0.054, 0.055–070, and ≥0.071; PM_2.5_ 24 h (μg/m^3^): ≤12.0, 12.1–35.4, and ≥35.5; SO_2_ 1 h (ppb): ≤35, 36–75, and >75; NO_2_ 1 h (ppb): ≤53, 54–100, and >100 [18]. Table A2 describes the responsible pollutant(s) when the AQI is unhealthy for sensitive groups or worse during the study period. AQI was determined from the maximum pollutant specific value across respective monitors for each pollutant. The average wind speed and temperature data were extracted from the National Oceanic and Atmospheric Administration (NOAA) monitoring station at Pittsburgh International Airport.

### Statistical Methods

Daily maximum temperatures (°F), average wind speeds (mph), and the AQI for criteria pollutants (ozone, PM_2.5_, SO_2_, and NO_2_) were described via measures of central tendency and dispersion. Differences in average temperature and wind speed across AQI categorical groups (good (AQI < 50), moderate (AQI 50–100), unhealthy (AQI > 100)) were compared via analysis of variance (ANOVA). The measures of the AQI at continuous scales were also evaluated in relation to the temperature and wind speed data via Spearman’s correlation coefficient. To assess the impact of outdoor air pollution exposure levels on daily exacerbation rates, we examined various exposure periods, including same-day exposure (lag_0_), exposure from the previous day (lag_1_), exposure six days prior (lag_5_), and average exposure over lag days 0–5 (lag_0–5_) similar to the lag structure approaches considered by Chen and colleagues (2024) [20]. Exacerbation rates were initially compared across same-day AQI levels categorized as good, moderate, and unhealthy. However, no significant differences were observed between days characterized by good and moderate air quality, as detailed in Table A3, Table A4, Table A5 and Table A6. Consequently, good, and moderate AQI levels were combined to evaluate the effects of outdoor air pollution exceedances (AQI > 100) on collective exacerbation rates and by age strata. Daily exacerbation rates for asthma, bronchitis, and COPD per 1000 residents were examined in relation to the AQI via the GLM procedure with a specified Poisson distribution (link = log). The ratio of the deviance to its degrees of freedom (deviance/df) was <1.5 across models, suggesting adequate fit and no evidence of overdispersion that would require quasi-Poisson or negative binomial models. The natural logarithm of the population size was included as an offset term in the analyses specific to the demographic group being analyzed. Forest plots were created to visually present exacerbation rate ratios and their corresponding 95% confidence interval estimates when AQI_lag_0_ > 100 vs. ≤100 overall and by age group. The impact of same-day outdoor air pollution exceedance on asthma exacerbation rates was further examined in relation to sex and insurance categories within two age strata: <18 years and 18–65 years. Forest plots were generated via R V4.4.2, while all remaining analyses were conducted via SPSS V29.0 (IBM, Armonk, NY, USA).

## 3. Results

Table 1 summarizes the overall study population, including distribution data for sex and insurance status across age groups according to the 2015–19 ACS 5-year estimates.

The Mon Valley has a total population of 117,545, with nearly 20% being under 18 years of age and another 20% being 65 years of age and older. The population was predominantly female (52.8%), with the proportion of females increasing with age: 48.7% in children, 52.3% in adults aged 18–64, and 57.9% in those aged 65 years and older. Coverage by public health insurance based on the ACS definition (coverage through Medicare, Medicaid, or other government-provided medical assistance programs) was high among children at 48.6% and 26.6% among adults aged 18–64 years and nearly universal among adults aged 65 years and older. Figure 1 shows the distribution of visit types for asthma (N = 675), bronchitis (N = 423), and COPD (N = 696) exacerbations during the study period. For asthma, the most common visit types were the emergency department followed by outpatient visits. Most bronchitis events occurred in outpatient setting followed by the ED. COPD exacerbations showed a more even distribution across settings, with the majority being outpatient visits with a notable share requiring hospital observation or admission.

Table 2 summarizes the AQI range, maximum daily temperature, and average daily wind speed during the study period. Most days had good (AQI < 50) or moderate (AQI 50–100) air quality.

However, during the study period, there were 37 days with air quality exceedances (AQI > 100): ozone exceedance occurred on 6 days, PM_2.5_ exceedances occurred on 17 days, and SO_2_ exceedances occurred on 16 days, with two of those days recording exceedances of both PM_2.5_ and SO_2_ (detailed in Table A1). NO_2_ levels did not exceed 53 ppb (1 h) during the study period, limiting the evaluation of this pollutant by the AQI categorically. The outdoor air temperature averaged 10 degrees Fahrenheit higher on days with moderate or unhealthy AQI levels than on those with good AQI levels (*p* < 0.05), as shown in Table 2. This temperature difference is further evidenced by a direct correlation between the AQI on a continuous scale and warmer temperatures (r_s_ = 0.302, *p* < 0.05).

Table 3 shows exacerbation rate ratios for asthma, bronchitis, and COPD, comparing days with AQI > 100 versus those with AQI ≤ 100 across different exposure periods: same-day exposure (lag_0_), exposure from the previous day (lag_1_), average exposure across lag days 0–5 (lag_0–5_), and exposure six days prior (lag_5_).

This analysis focused on overall AQI exceedances (>100), with PM_2.5_ and SO_2_ identified as the primary contributing pollutants. An association between exposure to unhealthy air quality and the risk of asthma exacerbations within the same day was identified (Table 3).

The rate ratio (RR) of asthma exacerbations was 1.42 times greater on days when the AQI was >100 than when it was ≤ 100 (95% CI: 1.05, 1.95; *p* = 0.025). A higher rate ratio was observed when PM_2.5_ and SO_2_ exceedances were considered individually (RR = 1.60; 95% CI: 1.06, 2.43; *p* = 0.026 for each). Elevated exposure levels (AQI > 100 versus AQI ≤ 100) on lag day 1, lag day 5 and lag days 1–5 were not related to the asthma exacerbation rate (*p* ≥ 0.05).

The rate ratio of bronchitis exacerbations was 1.76 times greater on lag day 5 (95% CI: 1.08, 2.93; *p* = 0.024) (Table 3). This relationship was not observed for same-day exposure or prior day exposure (lag day 1) for either PM_2.5_ or SO_2_. When evaluated specifically for each pollutant, the average exposure across lag days 0–5 did not exceed AQI >100.

Across the full cohort, no significant association was observed between elevated AQI levels (AQI > 100) and the risk of COPD exacerbation (Table 3). Similarly, individual pollutant components such as PM_2.5_ and SO_2_ did not show statistically significant associations with COPD exacerbation rates in the unadjusted analyses.

Figure 2 shows the age-dependent impact of outdoor air pollution exceedances on the rate of asthma exacerbations.

SO_2_ exceedances were associated with a greater than threefold increased rate of asthma exacerbations in children (RR = 3.33; 95% CI 1.45, 7.62; *p* = 0.004). The rate ratio of asthma exacerbations increased in adults aged 18–65 years when PM_2.5_ exceeded that threshold (RR = 1.71; 95% CI: 1.07, 2.74; *p* = 0.026). Similarly, the rate ratio of bronchitis exacerbations increased in adults aged 18–65 years when SO_2_ exceeded this threshold (RR = 2.09, 95% CI: 1.11, 3.83; *p* = 0.017) (Figure 3).

No significant effect of outdoor air quality on the rate ratios of asthma and bronchitis exacerbations was identified among adults aged 65 years and older. Furthermore, no significant association between same-day air quality and the percentage of COPD exacerbations was detected in the adult population (Figure 4).

Table 4 further examines asthma exacerbations to identify the subgroups most affected by outdoor air pollution.

Among children, the effect was most pronounced, particularly for females and those with public health insurance, who experienced significantly higher asthma exacerbation rates on AQI exceedance (vs. no exceedance) days. In children, SO_2_ exceedance (vs. no exceedance) corresponded to a fivefold increased rate of asthma exacerbations in females and a fourfold increased rate in those with public health insurance (RR = 5.05; 95% CI: 1.80, 14.19; *p* = 0.002 and RR = 4.01; 95% CI: 1.74, 9.24; *p* = 0.001, respectively). In contrast, among adults aged 18–65 years, SO_2_ exceedances (vs. not) increased the risk of asthma exacerbation twofold in males (RR = 2.15; 95% CI: 1.06, 4.37; *p* = 0.034) but not females.

## 4. Discussion

The study revealed that higher AQI levels were significantly associated with increased rates of asthma exacerbations, particularly on the same day of pollutant exposure. These findings underscore the disproportionate burden of air pollution among demographic groups and contribute to the broader discourse on environmental justice. These findings highlight the urgent need for targeted regulatory interventions and expanded access to healthcare services. The risk varies by pollutant; SO_2_ has emerged as a primary contributor to respiratory exacerbation, particularly asthma-related events among children, and bronchitis in adults aged 18–64 years. Children were disproportionately affected, with the highest impact observed among females and those covered by public health insurance.

This detailed analysis offers valuable insight into which demographic groups are most affected by poor air quality. Specifically, it elucidates the relationship between air pollution and respiratory health exacerbations, emphasizing the public health implications of outdoor air pollution. Our findings align with the literature, demonstrating a significant association between exposure to unhealthy air quality and increased respiratory morbidity. AQI exceedances, particularly of PM_2.5_ and SO_2,_ have emerged as the primary pollutants contributing to respiratory harm [22,23]. This study provides evidence of both immediate and delayed health effects associated with exposure to these pollutants.

The demographic distribution in our study highlights the broader issue of environmental health disparities. Nearly half of the children were covered by public health insurance, reflecting underlying socioeconomic and structural factors that increase vulnerability to air pollution [16]. Our findings indicate that females and publicly insured individuals, particularly children, are disproportionately affected by poor air quality, which is consistent with previous research documenting similar disparities in both exposure and health outcomes related to air pollution [18]. These results emphasize the unequal burden of air pollution on vulnerable populations, who experience higher rates of respiratory exacerbations in response to elevated pollutant levels. This underscores the need for targeted public health strategies and policy interventions to address persistent environmental health inequities.

Our analysis of healthcare utilization for asthma, bronchitis, and COPD exacerbations revealed notable differences in patterns of care. These variations may reflect differing acute responses to air pollution or disparities in healthcare access, echoing concerns in prior studies regarding the role of environmental factors in health service utilization. Despite findings from other studies that identified strong associations between air pollution and COPD-related hospital admissions, our data did not show similar findings [20,24]. This discrepancy may be due to recent shifts in care practices, including improved treatment adherence, increased home-based management, and a preference for outpatient care over hospitalization for acute asthma and COPD patients [25,26,27,28].

Our findings demonstrate a clear temporal association between air quality and respiratory health exacerbations, underscoring the urgency of interventions to mitigate exposure to harmful pollutants. This immediate effect of air pollution on asthma exacerbation rates was also demonstrated in a previous study documenting the acute impacts of PM_2.5_ and SO_2_ on respiratory health [29]. Additionally, our observation of delayed bronchitis exacerbations associated with SO_2_ exposure contributes to the growing body of evidence on the lagged health effects of air pollution [20].

AQI alerts are designed to protect residents by providing timely information about pollution levels, which enables individuals to take proactive steps to reduce their health risks. However, the effectiveness of these alerts depends on public awareness and individuals choosing to act on the warnings. This is evidenced by a recent study that demonstrated that AQI alerts are not sufficient, and additional actions are needed to add value to the public AQI alerts [30]. Similarly, a study conducted in the Mon Valley demonstrated that many residents impacted in the aftermath of U.S. Steel’s Clairton Coke Works industrial accident in 2018 were unaware of persistently elevated air pollution levels despite the release of AQI alerts [31]. The conclusion from these results is that effective notification systems must build community capacity by increasing awareness and supporting the development of self-management and advocacy skills in response to poor air quality. The CATCH Resident program was recently launched in the Dallas-Fort Worth Texas region to evaluate the effectiveness of community wide initiatives to improve air pollution knowledge and awareness [32]. Results from this program are not yet available.

A 2011 study emphasized the need for accurate measurement and monitoring of environmental health disparities to promote environmental justice and inform effective policy [33]. A 2023 study further reinforced the disproportionate burden imposed by vulnerable groups, showing that residents of environmental justice communities are more likely to experience worsened asthma control [34]. This increased risk is attributed to several intersecting factors, proximity to industrial facilities and traffic, limited healthcare access, and socioeconomic hardships that hinder effective disease management. These communities often face systemic barriers such as poor housing and a lack of clean air initiatives. Targeted public health interventions and environmental policies are essential to mitigate these inequities and improve respiratory outcomes.

While the national air quality has improved in recent years, recent large-scale studies have shown that adverse health effects occur at lower pollution levels than previously recognized. This contributed to the recent EPA revision of its annual PM_2.5_ standard from 12 µg/m^3^ to 9 µg/m^3^, reflecting a new scientific consensus and reinforcing the need for continued vigilance and regulation. Our study also highlights age-dependent differences in air pollution sensitivity among different age groups. We observed increased asthma and bronchitis exacerbation rates among children and adults aged 18–64 days with elevated SO_2_ and PM_2.5_ levels. These findings are consistent with the literature identifying these age groups as particularly susceptible. However, we did not observe a statistically significant impact on respiratory exacerbations in adults aged 65 and older, a result that diverges from those of some prior studies and warrants further investigation into age-related vulnerabilities [35].

These results suggest that the AQI can be a useful public health tool, helping residents minimize exposure and reduce health risks on days with poor air quality. More importantly, our study also reinforces the need for policies that prioritize primary prevention by decreasing exposure to air pollution in vulnerable populations. Such policies have been effective in improving national air quality and certain health outcomes over the past several decades. However, recent large-scale studies have shown that adverse health effects occur at lower pollution levels than previously recognized. This contributed to the recent EPA revision of its annual PM_2.5_ standard from 12 µg/m^3^ to 9 µg/m^3^, reflecting a new scientific consensus and reinforcing the need for continued vigilance and regulation. Moreover, public education, continuous monitoring, and the integration of AQI data into clinical practice could strengthen both personal prevention strategies and systemic healthcare planning to manage respiratory disease more effectively.

As with any study, there are limitations. Our analysis focused only on short-term effects and did not examine long-term effects of exposure to elevated levels of air pollution. We designed the study this way because AQI is typically interpreted as an acute exposure method. We acknowledge that AQI can also be aggregated for cumulative assessment and that this should be examined in future long-term studies. Furthermore, we used several EPA-grade reference monitors from distinct locations within Allegheny County to perform the analyses. Because of this small air monitor network, we relied on regional pollution data rather than hyper-local conditions. Therefore, we were unable to capture local-scale spatial variations that may have occurred for various pollutants. Due to the limited nature of our dataset, confounders such as socioeconomic status, race and ethnicity, disease severity, use of controller therapy, concomitant diseases, and exposure to other triggers such as infection, tobacco smoke and allergens could not be examined. Visit severity was recorded as the highest level of care received during the encounter. For example, if a patient visited a primary care provider for an acute attack, it was referred to as ED, and subsequently admitted, only hospitalization was considered. While this approach minimizes duplication, it omits data on immediate care encounters. Also, the measured outcomes are only one piece of a larger picture that describes the burden of respiratory diseases. Future studies should consider additional endpoints (i.e., symptoms, rescue medication use, impact on school/work absences, physical activity, and sleep) to obtain a more comprehensive picture of how respiratory disease control is exacerbated by air pollutants. The study did not assess participants’ awareness, understanding and any actions taken to protect their health in response to AQI alerts. The study period (2018–2020) was relatively short and may limit the robustness of the findings. We selected this period because AHN changed their database system at the beginning of 2018, and it would have been extremely difficult to merge data from prior years that were collected under another system. Additionally, we selected early 2020 as the cut-off date to eliminate COVID infections as a confounder. The study did not adjust for additional temporal factors such as humidity, holidays, or long term/seasonal trends; given the relatively short study period and the limited number of AQI exceedance days, we prioritized a parsimonious model structure. That said, post hoc analyses adjusting for calendar season did not alter the significance of the findings described in Table 3 and Table 4 and Figure 2, Figure 3 and Figure 4, providing assurance that our results are robust to seasonal adjustment. An exception was observed for bronchitis event rates: the overall RR for SO_2_ lag5 decreased from 1.76 to 1.52 (95% CI 0.92, 2.52, *p* = 1.03) in Table 3 and for SO_2_ lag0 among adults aged 18–64 years decreased from 2.09 to 1.69 (95% CI 0.92, 3.11, *p* = 0.092). These limitations highlight areas for future research to characterize the full spectrum of the health impacts of air pollution better. Future longitudinal studies should examine the health impacts of both short-term and long-term exposure to air pollution and incorporate a thorough characterization of patient demographics, disease severity and control, concomitant conditions, disease triggers and understanding and use of AQI alerts. Additionally, the time period should be significantly expanded, and consideration should be given to determine hyper-local air pollution conditions by equipping participants with low-cost, accessible air pollution monitors.

## 5. Conclusions

In conclusion, despite several methodological limitations, this study demonstrates that poor air quality, particularly PM_2.5_ and SO_2_, is strongly associated with acute respiratory morbidity, with immediate effects on asthma and delayed effects on bronchitis and particularly among vulnerable populations such as children, women, and individuals with public insurance, underscoring persistent environmental health disparities. Our findings highlight the potential value of the AQI as a predictive public health tool and justify the need for future well-designed longitudinal studies to determine if AQI alerts can inform clinical practice and healthcare planning. Such studies should include a longer time period, better characterization of demographics and confounders, respiratory disease severity and controller therapy, concomitant medical conditions, and use of hyper-local air pollution measurements. Collectively, targeted interventions such as AQI based alerts, stricter regulation of industrial emissions, and expanded access to preventative care for vulnerable populations are essential to reduce the health burden of air pollution and inequities in respiratory outcomes. This is in line with the global health priorities advocated by organizations such as the World Health Organization.

## Figures and Tables

**Figure 1 ijerph-22-01493-f001:**
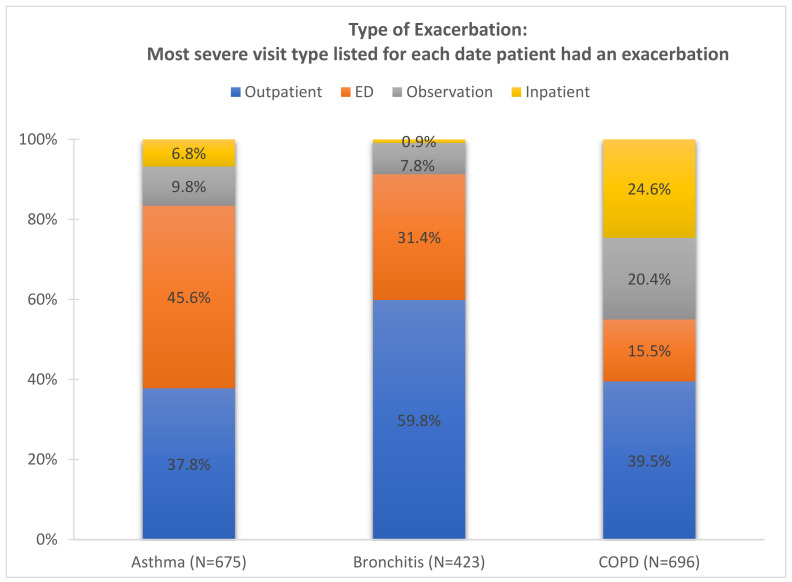
Exacerbation type assigned as most severe on date for each condition within patient documented in Allegheny Health Network data from Jan 2018–Feb 2020 (790 days in population of 117,545 residents).

**Figure 2 ijerph-22-01493-f002:**
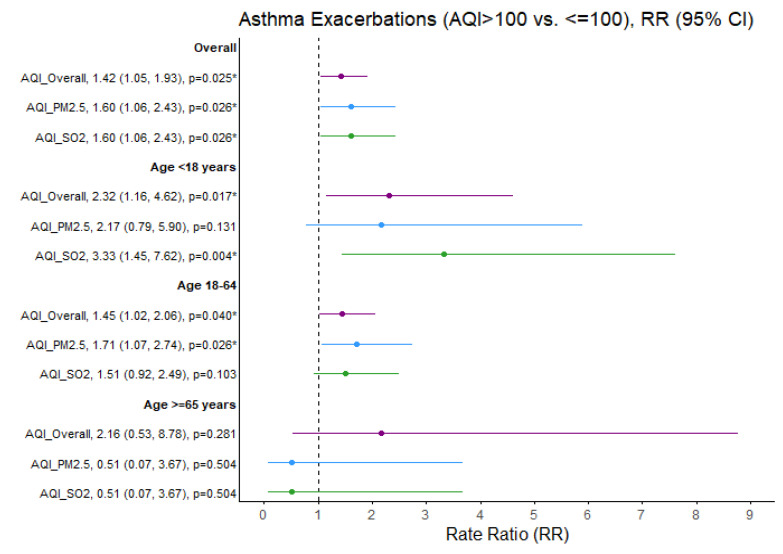
Asthma exacerbation rate ratios (RRs) comparing air quality days (AQI > 100 vs. ≤100) overall and specific to PM_2.5_ and SO_2_ pollutants. * *p* < 0.05, exacerbation rate ratios (95% CI) estimates and significance based on GLM Poisson regression. Ozone and NO2 not presented due to limited days when AQI > 100 (N = 6 and N = 0 days, respectively). Purple = Overall AQI (>100 vs. ≤100), Blue = PM_2.5_ AQI (>100 vs. ≤100), Green = SO_2_ AQI (>100 vs. ≤100).

**Figure 3 ijerph-22-01493-f003:**
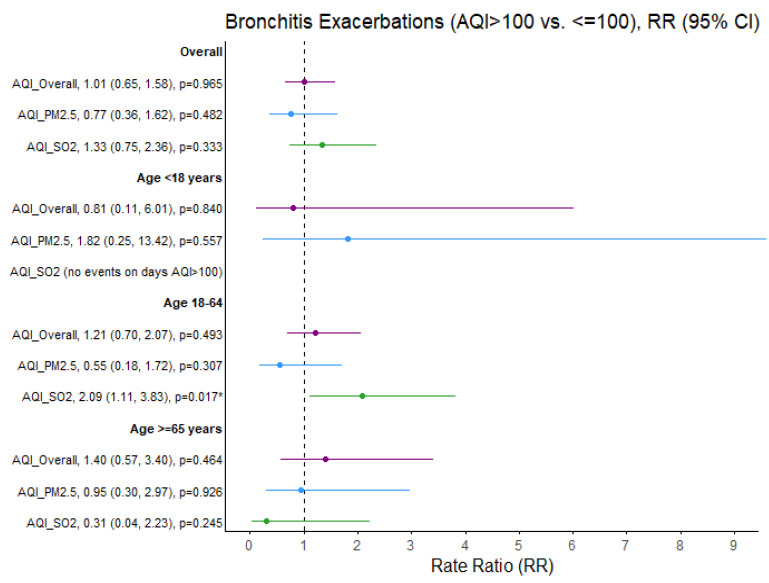
Bronchitis exacerbation rate ratios (RRs) comparing air quality days (AQI > 100 vs. ≤100) overall and specific to PM_2.5_ and SO_2_ pollutants. * *p* < 0.05, exacerbation rate ratios (95% CI) estimates and significance based on GLM Poisson regression. Ozone and NO2 not presented due to limited days when AQI > 100 (N = 6 and N = 0 days, respectively). Purple = Overall AQI (>100 vs. ≤100), Blue = PM_2.5_ AQI (>100 vs. ≤100), Green = SO_2_ AQI (>100 vs. ≤100).

**Figure 4 ijerph-22-01493-f004:**
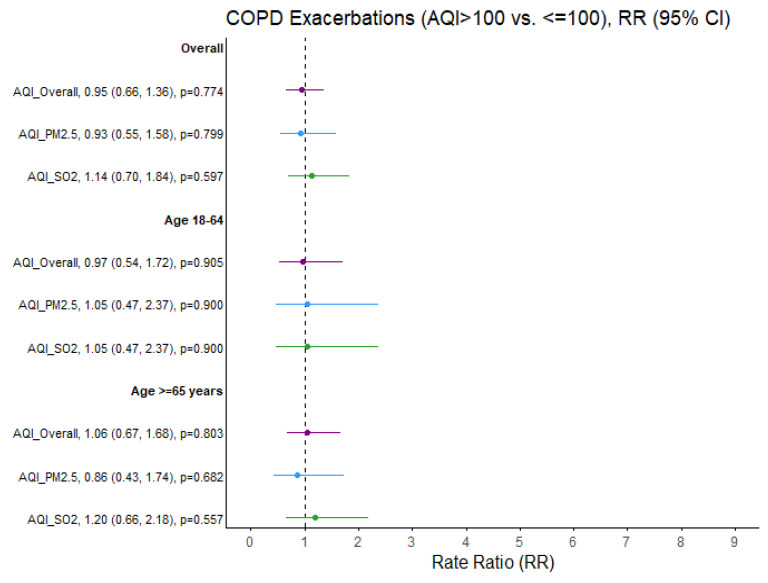
COPD exacerbation rate ratios (RRs) comparing air quality days (AQI > 100 vs. ≤100) overall and specific to PM_2.5_ and SO_2_ pollutants. Ozone and NO2 not presented due to limited days when AQI > 100 (N = 6 and N = 0 days, respectively). Purple = Overall AQI (>100 vs. ≤100), Blue = PM_2.5_ AQI (>100 vs. ≤100), Green = SO_2_ AQI (>100 vs. ≤100).

**Table 1 ijerph-22-01493-t001:** Description of study populations (N = 117,545 ^a^) overall and by age group (January 2018–February 2020).

	Overall	Age Group:		
		<18 Years	18–64 Years	≥65 Years
	N = 117,545	N = 23,050	N = 70,280	N = 24,215
Sex:				
Male	55,531 (47.2%)	11,836 (51.3%)	33,502 (47.7%)	10,193 (42.1%)
Female	62,014 (52.8%)	11,214 (48.7%)	36,778 (52.3%)	14,022 (57.9%)
Insurance:				
Public	53,151 (45.2%)	11,212 (48.6%)	18,711 (26.6%)	23,592 (97.4%)
Private	64,394 (54.8%)	11,838 (51.4%)	51,569 (73.4%)	623 (2.6%)

^a^ Source: U.S. Census Bureau, 2015–2019 American Community Survey (ACS) 5-Year Estimates [21].

**Table 2 ijerph-22-01493-t002:** Air quality index (AQI), maximum daily temperature, and average daily wind speed distributions across the 790-day study period (Jan 2018–Feb 2020).

	Overall		Air Quality Index (790 Days)		
	Mean (SD)	Median [IQR]	Good (AQI < 50)	Moderate (AQI 50–100)	Unhealthy (AQI >100)
**AQI:**			#Days (%)	#Days (%)	#Days (%)
Overall	53.1 (23.2)	48.8 [36.5, 64.1]	416 (52.7%)	337 (42.7%)	37 (4.7%)
Ozone	34.7 (16.5)	32.4 [24.1, 41.7]	686 (90.1%)	69 (9.1%)	6 (0.8%)
PM_2.5_	48.1 (22.8)	45.8 [30.8, 61.0]	443 (56.1%)	330 (41.8%)	17 (2.2%)
SO_2_	23.6 (24.6)	15.7 [4.3, 34.3]	693 (87.7%)	80 (10.1%)	17 (2.2%)
NO_2_	18.8 (7.5)	17.8 [13.4, 23.6]	790 (100%)	----	----
			Mean (SD)	Mean (SD)	Mean (SD)
Max Temp (F)	60.1 (19.8)	61.0 [42.0, 79.0]	55.1 (19.1)	66.1 (19.3)	66.1 (19.3) *
Wind Speed (mph)	16.7 (6.8)	16.0 [11.5, 21.0]	17.9 (7.03)	15.1 (6.2)	17.6 (6.6) *

Valid % reported in table (excludes days with missing pollution metric in calculation). Temp = maximum daily temperature. Wind speed (daily average wind speed: recorded in tenths of meters per second NOAA and converted to mph by multiplying by 2.2369. * *p* < 0.05, significant difference in average temperature across the three AQI overall categories based on ANOVA.

**Table 3 ijerph-22-01493-t003:** Exacerbation rate ratios (RRs) comparing AQI (>100 vs. ≤100) on days LAG_0_, LAG_1_, LAG_5_, and cumulative exposure (AQI meanLAG_0–5_) ^a^, N = 790 days.

RR Comparing AQI (>100 vs. ≤100)	Daily Exacerbation Rate Ratios (95% CI)		
	Asthma	Bronchitis	COPD
**AQI Overall**	RR (95% CI)	RR (95% CI)	RR (95% CI)
LAG_0_	1.42 (1.05, 1.93) *	1.01 (0.65, 1.58)	0.95 (0.66, 1.36)
LAG_1_	0.79 (0.54, 1.17)	0.93 (0.59, 1.47)	0.95 (0.67, 1.36)
LAG_5_	1.04 (0.73, 1.47)	1.11 (0.72, 1.70)	1.17 (0.84, 1.62)
meanLAG_0–5_ ^a^	0.78 (0.35, 1.74)	0.83 (0.31, 2.22)	1.27 (0.68, 2.36)
**AQI_PM2.5** ^b^			
LAG_0_	1.60 (1.06, 2.43) *	0.77 (0.36, 1.62)	0.93 (0.55, 1.58)
LAG_1_	0.96 (0.57, 1.63)	0.99 (0.51, 1.91)	0.86 (0.50, 1.50)
LAG_5_	0.89 (0.51, 1.54)	0.76 (0.36, 1.60)	1.41 (0.91, 2.17)
**AQI_SO_2_** ^b^			
LAG_0_	1.60 (1.06, 2.43) *	1.33 (0.75, 2.36)	1.14 (0.70, 1.84)
LAG_1_	0.68 (0.37, 1.28)	1.10 (0.59, 2.06)	1.21 (0.76, 1.93)
LAG_5_	1.24 (0.78, 1.98)	1.76 (1.08, 2.93) *	0.93 (0.55, 1.58)

* *p* < 0.05, exacerbation rate ratios (95% CI) estimates and significance based on GLM Poisson regression. ^a^ AQI overall meanLAG_0–5_ distribution: 347 good, 434 moderate, 9 unhealthy for sensitive groups which were all attributable to PM_2.5_. ^b^ meanLAG_0–5_ not presented for ozone and NO_2_ due to limited days when AQI > 100 (N = 6 and N = 0 days, respectively).

**Table 4 ijerph-22-01493-t004:** Asthma exacerbation rate ratios (RR) comparing AQI_LAG_0_ (>100 vs. ≤100) evaluated overall and within age, gender, and insurance stratums, n = 790 days (Jan 2018–Feb 2020).

	Asthma Exacerbation Rate Comparing AQI_ LAG_0_ > 100 vs. ≤100:			
	AQI Overall RR (95% CI)	AQI Ozone RR (95% CI)	AQI PM2.5 RR (95% CI)	AQI SO_2_ RR (95% CI)
** Stratum: **				
**Overall**	1.42 (1.05, 1.93) *	1.73 (0.56, 5.37)	1.60 (1.06, 2.43) *	1.60 (1.06, 2.43) *
**Age < 18 years**	2.32 (1.16, 4.62) *	1.50 (0.21, 10.76)	2.17 (0.79,5.90)	3.33 (1.45, 7.62) *
Sex:				
Male	1.85 (0.67, 5.15)	----	3.03 (0.94, 9.75)	1.98 (0.48, 8.14)
Female	2.91 (1.14, 7.42) *	3.40 (0.50, 24.79)	1.17 (0.16, 8.49)	5.05 (1.80, 14.19) *
Insurance:				
Public	2.47 (1.18, 5.14) *	1.80 (0.25, 12.94)	1.92 (0.61, 6.10)	4.01 (1.74, 9.24) *
Private ^a^	1.57 (0.21, 11.97)	----	----	----
**Age 18–64 years**	1.45 (1.02, 2.06) *	0.52 (0.13, 2.10)	1.71 (1.07, 2.74) *	1.51 (0.92, 2.49)
Sex:				
Male	1.61 (0.92, 2.84)	----	1.87 (0.88, 3.99)	2.15 (1.06, 4.37) *
Female	1.36 (0.86, 2.13)	0.81 (0.20, 3.26)	1.62 (0.89, 2.95)	1.17 (0.58, 2.35)
Insurance:				
Public	1.44 (0.90, 2.29)	0.93 (0.23, 3.72)	1.64 (0.87, 3.07)	1.30 (0.64, 2.62)
Private	1.46 (0.85, 2.51)	----	1.81 (0.89, 3.67)	1.81 (0.89, 3.67)

* *p* < 0.05. AQI > 100 = Unhealthy for sensitive people or worse; AQI ≤ 100 = Good to moderate. Exacerbation is defined by primary Dx of asthma for outpatient visit, ED visit, hospital observation, or IP stay. Exacerbation rate ratios (95% CI) estimates and significance based on GLM Poisson regression. ^a^ In patients < 18 years of age with private insurance, there were 14 asthma exacerbations of which only one occurred on an unhealthy AQI day (inadequate data to evaluate by each specific pollution metric). Age ≥ 65 not evaluated by demographics characteristics due to limited number of exacerbations in select strata.

## Data Availability

The original contributions presented in this study are included in the article. Further inquiries can be directed at the corresponding author(s).

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
