# Peer review of "Air Quality Index as a Predictor of Respiratory Morbidity in At-Risk Populations"

_ijerph, 2025, doi:10.3390/ijerph22101493_

Round 1

Reviewer 1 Report

Comments and Suggestions for Authors

Dear Authors,

Below are my comments regarding the manuscript entitled "Air Quality Index as a Predictor of Respiratory Morbidity in At-Risk Populations":

In Abstract

  • It does not clearly articulate the study’s original contribution to the existing literature. I recommend that the authors revise the abstract to explicitly highlight the novel aspects of their research and how it advances current knowledge in the field.

In Introduction:

  • While the manuscript states that the Air Quality Index (AQI) is used in the United States for air quality monitoring and public health advisories, it is important to note that the AQI is also widely used or adapted in various countries around the world. Therefore, this statement could be revised to reflect its broader, global relevance as an air quality monitoring tool.
  • The authors present a relevant and timely study on the disproportionate impact of air pollution across demographic groups. However, the manuscript would benefit from a clearer articulation of how this research differs from previous studies in this area. Specifically, highlighting the unique aspects of the data, methodology, or focus population would help emphasize the originality and added value of the study.

In Methods

  • The Methods section could be strengthened by including more detailed information on the geographic characteristics of the study area. Additionally, presenting the data collection locations on a map would enhance the clarity of the study design and provide helpful context for interpreting the findings.

In Results

  • In Figure 1, the abbreviation "ED" is used without clarification. It is recommended that the authors explicitly define "ED" as "Emergency Department" either in the figure legend or at its first mention in the main text to ensure clarity for all readers.
  • The numerical data presented in Lines 150–154 appears to largely replicate the information already shown in Figure 1. To improve the readability and avoid redundancy, the authors may consider summarizing the overall trends or key takeaways from the figure in the text, rather than repeating the specific percentages. A more interpretive or comparative summary could enhance the clarity and impact of the Results section.

In Discussion and Conclusion

  • It is recommended that the authors present the Discussion and Conclusion sections under separate headings. Separating these sections can improve the structural clarity of the manuscript by allowing a more focused interpretation of the findings in the Discussion, followed by a concise summary of the main implications, and recommendations in the Conclusion. This distinction can help readers more easily differentiate between in-depth analysis and final remarks.

Author Response

Reviewer 1 Comments and Responses

In Abstract:

  • It does not clearly articulate the study’s original contribution to the existing literature. I recommend that the authors revise the abstract to explicitly highlight the novel aspects of their research and how it advances current knowledge in the field.

We agree and have modified the abstract to clearly state the novel aspects of this study, lines 13-27.  Specifically, we now state that recent studies have linked air pollution in this region to asthma exacerbations. Our study extends this work by examining a broader set of outcomes, three highly relevant respiratory conditions, across both age and insurance groups. This approach allows us to assess not only the overall health impacts of unhealthy air quality but also differential effects among vulnerable populations. 

In Introduction:

  • While the manuscript states that the Air Quality Index (AQI) is used in the United States for air quality monitoring and public health advisories, it is important to note that the AQI is also widely used or adapted in various countries around the world. Therefore, this statement could be revised to reflect its broader, global relevance as an air quality monitoring tool.

We agree and have modified the introduction as suggested, lines 33-35.  Additionally, two references have been added to support this modification.

  • The authors present a relevant and timely study on the disproportionate impact of air pollution across demographic groups. However, the manuscript would benefit from a clearer articulation of how this research differs from previous studies in this area. Specifically, highlighting the unique aspects of the data, methodology, or focus population would help emphasize the originality and added value of the study.

We agree and have clarified the introduction lines 66-70.  As with the revised abstract, we now state that recent studies have linked air pollution in this region to asthma exacerbations. Our study extends this work by examining a broader set of outcomes, three highly relevant respiratory conditions, across both age and insurance groups. This approach allows us to assess not only the overall health impacts of unhealthy air quality but also differential effects among vulnerable populations.

In Methods

  • The Methods section could be strengthened by including more detailed information on the geographic characteristics of the study area. Additionally, presenting the data collection locations on a map would enhance the clarity of the study design and provide helpful context for interpreting the findings.

We agree and have added a map as suggested (Supplemental Figure A1, inserted on line 395. Additionally, the methods section was updated with more detailed information on the geographic characteristics of the study area, lines 80-85.

In Results

  • In Figure 1, the abbreviation "ED" is used without clarification. It is recommended that the authors explicitly define "ED" as "Emergency Department" either in the figure legend or at its first mention in the main text to ensure clarity for all readers.

This change has been made as suggested. Additionally, changed “emergency department” later mentioned throughout the manuscript to “ED” for consistency.

  • The numerical data presented in Lines 150–154 appears to largely replicate the information already shown in Figure 1. To improve the readability and avoid redundancy, the authors may consider summarizing the overall trends or key takeaways from the figure in the text, rather than repeating the specific percentages. A more interpretive or comparative summary could enhance the clarity and impact of the Results section.

Response: We agree and have revised the text accordingly. Lines 138-142 (previously 150-154).

In Discussion and Conclusion

  • It is recommended that the authors present the Discussion and Conclusion sections under separate headings. Separating these sections can improve the structural clarity of the manuscript by allowing a more focused interpretation of the findings in the Discussion, followed by a concise summary of the main implications, and recommendations in the Conclusion. This distinction can help readers more easily differentiate between in-depth analysis and final remarks.

We agree and have made this change.  A header titled “5. Conclusion” has been added to the final section of the manuscript on line 328. Additionally, we rephrased the conclusion to be more concise. Lines 329-336.

Reviewer 2 Report

Comments and Suggestions for Authors

Introduction:

The text mentions in a rather generic way the association between air pollution and respiratory health, but it does not elaborate on the pathophysiological mechanisms nor distinguish between short- and long-term effects. In addition, it fails to situate the most susceptible groups (children, the elderly, individuals with comorbidities) based on established literature.

The introduction discusses the AQI as a monitoring tool but does not explain why it is used as the main analytical variable. Would it not be more robust to directly employ pollutant concentrations?

The introduction shows that multiple studies have already associated air pollution with asthma exacerbations in the Mon Valley. What does this study add that is new?

The text does not anticipate the methodological approach. In epidemiological studies, at least a brief indication of the statistical method employed is expected. Such an early reference would demonstrate scientific rigor and help the reader understand how the hypothesis will be tested.

Methods

Please clarify the inclusion/exclusion criteria for patients (e.g., minimum residence requirements within ZIP codes, observation period per patient). Also specify the exact ICD-10 codes used. In addition, indicate how repeat visits for the same patient were handled (e.g., was a washout period applied to avoid double counting of the same exacerbation episode?) and explain whether visit categories such as observation and hospital admission were considered separately or could overlap. This will strengthen reproducibility and avoid ambiguity.

Describe in detail how daily AQI was derived: was it based on the maximum of pollutant-specific sub-indices or an average across monitors? How were multiple monitoring sites combined, and how were missing data handled? Providing a map of monitor locations and their distance to the study ZIP codes would strengthen confidence in the exposure assignment.

Only temperature and wind speed are included as covariates. In short-term air pollution epidemiology, it is standard practice to also control for humidity, day of the week, holidays, seasonal and long-term trends (often with spline functions), and potential autocorrelation in the residuals. Please clarify whether these factors were considered in the analysis and, if not, provide a justification.

Please justify these lag selections with supporting references.

Please report the link function, check for overdispersion, and indicate whether quasi-Poisson or negative binomial models were considered. Describe how offsets were applied to stratified populations, and whether robust standard errors were used.

Add methodological citations to support the analytic framework.

Results:

The study period (2018–2020) is relatively short for time-series analyses and may limit the robustness of the findings. It would be important to justify this choice more clearly. In addition, the analysis should at least consider seasonality, for example by examining differences across seasons of the year, since both air pollution patterns and respiratory outcomes are known to vary seasonally.

Author Response

Reviewer 2 Comments and Responses

The text mentions in a rather generic way the association between air pollution and respiratory health, but it does not elaborate on the pathophysiological mechanisms nor distinguish between short- and long-term effects. In addition, it fails to situate the most susceptible groups (children, the elderly, individuals with comorbidities) based on established literature.

We acknowledge that our manuscript provides a succinct background on the relationship between air pollution and respiratory health using references 13–19. The revised discussion of study limitations now addresses that we focused on short-term effects and not long-term effects.  While detailed discussion of pathophysiological mechanisms is well established in the literature and considered beyond the scope of our study, we agree that it is important to emphasize populations shown to be most vulnerable, including children, the elderly, and individuals with pre-existing comorbidities. These groups are consistently identified in prior studies as disproportionately affected by both short- and long-term exposures. In our study, we specifically examined vulnerability within the Mon Valley by stratifying analyses by age and insurance status, as this setting is characterized by both elevated pollution exposures and a high prevalence of socioeconomically disadvantaged groups.  We also included in our examination exacerbation rates for asthma, bronchitis, and COPD.

The introduction discusses the AQI as a monitoring tool but does not explain why it is used as the main analytical variable. Would it not be more robust to directly employ pollutant concentrations?

We appreciate the reviewer’s question regarding our use of the Air Quality Index (AQI) as the primary analytic variable rather than raw pollutant concentrations. The AQI is the standard metric for air quality monitoring and public health communication in the United States and aligns with how risk is conveyed in real-world settings, thereby enhancing the translational relevance of our findings. It is widely disseminated to the public, including through weather applications (e.g., iPhone alerts), because it provides a standardized way to translate multiple pollutants with different units of measure and health-based thresholds on a common scale. This standardization allows for meaningful comparisons across pollutants and straightforward interpretation of health risks.  Lines 275-285 of our manuscript further  address the value of AQI. This is consistent with our reference #7 (Horn, 2024), which notes that the AQI not only standardized reporting within the U.S. but also provided a framework for international regulatory agencies seeking to improve public health communication.

The introduction shows that multiple studies have already associated air pollution with asthma exacerbations in the Mon Valley. What does this study add that is new?

The Mon Valley consistently reports some of the poorest air quality in the United States, particularly for PMâ‚‚.â‚… and SOâ‚‚, pollutants well documented to exacerbate respiratory conditions. While prior studies have linked air pollution in this region to asthma exacerbations, our study extends this work by examining a broader set of outcomes, three highly relevant respiratory conditions, across both age and insurance groups. This approach allows us to assess not only the overall health impacts of unhealthy air quality but also differential effects among vulnerable populations, thereby providing new insights into health disparities within a region of persistently elevated exposures.

The text does not anticipate the methodological approach. In epidemiological studies, at least a brief indication of the statistical method employed is expected. Such an early reference would demonstrate scientific rigor and help the reader understand how the hypothesis will be tested.

We appreciate the reviewer’s suggestion regarding early indication of statistical methods. In our view, these methods are widely established in epidemiological research, and we sought to present them concisely in the dedicated Methods section (lines 106–127) rather than in the Introduction. For clarity, we note here that group differences in temperature and wind speed across AQI categories (good: <50, moderate: 50–100, unhealthy: >100) were assessed using analysis of variance (ANOVA). Associations between continuous AQI values and meteorological factors were evaluated using Spearman’s correlation. Daily exacerbation rates for asthma, bronchitis, and COPD per 1,000 residents were then examined in relation to AQI using generalized linear models (GLM) with a Poisson distribution. We believe this structured presentation demonstrates rigor while keeping the Introduction focused on the scientific context rather than methods.

Methods

Please clarify the inclusion/exclusion criteria for patients (e.g., minimum residence requirements within ZIP codes, observation period per patient). Also specify the exact ICD-10 codes used. In addition, indicate how repeat visits for the same patient were handled (e.g., was a washout period applied to avoid double counting of the same exacerbation episode?) and explain whether visit categories such as observation and hospital admission were considered separately or could overlap. This will strengthen reproducibility and avoid ambiguity.

In this ecological study, any exacerbation outcomes defined by respective ICD-10 codes for patients residing in the 11 ZIP codes listed in Supplemental Table A1 were included. Address-level information was not available from the hospital, so we were unable to apply minimum residential distance criteria. We agree with the reviewer’s point and have revised the text (lines 86–94) to state that only the most severe event per patient was recorded on a given date, thereby preventing overlap (e.g., if both observation and hospital admission occurred, only the admission was counted).  This is also stated in the discussion section.  In addition, we have now included lines 87-94 to describe the exact ICD-10 codes for each condition.

Describe in detail how daily AQI was derived: was it based on the maximum of pollutant-specific sub-indices or an average across monitors? How were multiple monitoring sites combined, and how were missing data handled? Providing a map of monitor locations and their distance to the study ZIP codes would strengthen confidence in the exposure assignment.

We agree with reviewer and have added additional description to line 103-105 “AQI was determined from the maximum pollutant specific value across respective monitors for each pollutant.”

Only temperature and wind speed are included as covariates. In short-term air pollution epidemiology, it is standard practice to also control for humidity, day of the week, holidays, seasonal and long-term trends (often with spline functions), and potential autocorrelation in the residuals. Please clarify whether these factors were considered in the analysis and, if not, provide a justification.

We agree that the absence of these covariates is an important consideration and have added this point to the Limitations section of the manuscript (lines 312-327). “The study did not adjust for additional temporal factors such as humidity, holidays, or long term/seasonal trends; given the relatively short study period and the limited number of AQI exceedance days, we prioritized a parsimonious model structure.”

Please justify these lag selections with supporting references.

We have added the following to the sentence on lines 111-115:  “Similar to the lag structure approaches considered by Chen and colleagues (2024) [21]. ” 

Specifically: To assess the impact of outdoor air pollution exposure levels on daily exacerbation rates, we examined various exposure periods, including same-day exposure (lag0), exposure from the previous day (lag1), exposure six days prior (lag5), and average exposure over lag days 0-5 (lag0-5), similar to the lag structure approaches considered by Chen and colleagues (2024) [21].

Please report the link function, check for overdispersion, and indicate whether quasi-Poisson or negative binomial models were considered. Describe how offsets were applied to stratified populations, and whether robust standard errors were used.

We have added the following sentence to statistical methods section lines 120-127 to address the reviewer’s concern:  “The ratio of the deviance to its degrees of freedom (deviance/df) was consistently <1.5 across models, suggesting adequate fit and no evidence of overdispersion that would require quasi-Poisson or negative binomial models.”  Further link=log was added to the prior sentence describing specification of Poisson distribution.

In the stratified population analyses the natural logarithm of the population size was included as an offset term in the analyses specific to the demographic group being analyzed as described in the statistical methods section.

Add methodological citations to support the analytic framework.

We appreciate the suggestion to add methodological citations to support the analytic framework. As the analytical approach relied on widely accepted statistical techniques (e.g., ANOVA, Spearman’s correlation coefficients, and Poisson regression), we did not initially feel it necessary to provide additional citations for these standard methods.

Results:

The study period (2018–2020) is relatively short for time-series analyses and may limit the robustness of the findings. It would be important to justify this choice more clearly. In addition, the analysis should at least consider seasonality, for example by examining differences across seasons of the year, since both air pollution patterns and respiratory outcomes are known to vary seasonally.

We appreciate the reviewer’s concern and have conducted post-hoc analyses including calendar season in the models to evaluate its impact on the significance of our findings presented in Tables 3 and 4 and Figures 2–4. To address this point in the manuscript, we added the following sentence to the limitations section of the discussion lines 322-327: “That said, post-hoc analyses adjusting for calendar season did not alter the significance of the findings described in Tables 3 and 4 and Figures 2–4, providing assurance that our results are robust to seasonal adjustment.  An exception was observed for bronchitis event rates: the overall RR for SO2 lag5 decreased from 1.76 to 1.52 (95% CI 0.92, 2.52, p=1.03) in Table 3 and for SO2 lag0 among adults aged 18-64 years decreased from 2.09 to 1.69 (95% CI 0.92, 3.11, p=.092).”

Reviewer 3 Report

Comments and Suggestions for Authors

The present study provides evidence about the importance of the AQI as both a surveillance tool and a mechanism for public health intervention. Emphasizing that poor air quality significantly affects respiratory health, particularly among vulnerable populations

This is an original and very well-designed study. The methodology used allows a clear correlation between the effect of air quality on respiratory health.

I only have one question: Why didn't they evaluate PM10 concentrations? These air pollutant concentrations, could be directly associated with exacerbations in asthmatic subjects.

Author Response

Reviewer 3 Comments and Responses

I only have one question: Why didn't they evaluate PM10 concentrations? These air pollutant concentrations could be directly associated with exacerbations in asthmatic subjects.

The revised methods now state that PM10 was not included in our analyses because levels have historically been well below regulatory thresholds and provides a reference to this data.  https://www.epa.gov/outdoor-air-quality-data/download-daily-data

Reviewer 4 Report

Comments and Suggestions for Authors

Authors:

  1. I suggest mentioning the unfortunate history of the Donora pollution episode in describing past studies in the Mon Valley.
  2. What are the sources of SO2 in the region? 
  3. While use of the AQI as the exposure metric is useful, does the AQI system itself have impact oni the consequences of air pollution exposure?  Take a look at the relatively limited literature on this point.
  4. Given the very extensive literature on ambient air pollution and respiratory morbidity, provide a clear statement on the contribution of this paper. 

Author Response

Reviewer 4 Comments and Responses

Authors:

  1. I suggest mentioning the unfortunate history of the Donora pollution episode in describing past studies in the Mon Valley.

We agree and have added text and a relevant reference to the introduction discussing the Donora Smog event on lines 45-47.

  1. What are the sources of SO2 in the region?

The revised introduction now states that US Steel’s Clairton Coke Works is the largest producer of SO2 in the region and includes a reference reflecting this, line 43.

  1. While use of the AQI as the exposure metric is useful, does the AQI system itself have impact oni the consequences of air pollution exposure?  Take a look at the relatively limited literature on this point.

We have added to the discussion addressing this issue and included an appropriate reference, lines 294-302.

  1. Given the very extensive literature on ambient air pollution and respiratory morbidity, provide a clear statement on the contribution of this paper. 

While prior studies have linked air pollution in this region to asthma exacerbations, our study extends this work by examining a broader set of outcomes, three highly relevant respiratory conditions, across both age and insurance groups. This approach allows us to assess not only the overall health impacts of unhealthy air quality but also differential effects among vulnerable populations, thereby providing new insights into health disparities within a region of persistently elevated exposures.

Round 2

Reviewer 1 Report

Comments and Suggestions for Authors

The authors have satisfactorily addressed all the previously raised concerns. At this stage, the only part of the manuscript that requires further improvement is the Conclusion section. Specifically, it would benefit from a more detailed discussion of the study's limitations and suggestions for future research

Author Response

Response uploaded as a WORD doc attached.

Reviewer 2 Report

Comments and Suggestions for Authors

The authors have addressed all my comments and made the necessary changes.

Author Response

We have  have addressed all of the reviewers comments and made the necessary changes.